# Robot Fine-Tuning Made Easy: Pre-Training Rewards and Policies for Autonomous Real-World Reinforcement Learning

**Abstract:** The pre-train and fine-tune approach in machine learning has been highly successful across various domains, enabling rapid task learning by utilizing existing data and pre-trained models from the internet. We seek to apply this approach to robotic reinforcement learning, allowing robots to learn new tasks with minimal human involvement by leveraging online resources. We introduce ROBOFUME, a reset-free fine-tuning system that pre-trains a versatile manipulation policy from diverse prior experience datasets and autonomously learns a target task with minimal human input. In real-world robot manipulation tasks, our method can incorporate data from an external robot dataset and improve performance on a target task in as little as 3 hours of autonomous real-world experience. We also evaluate our method against various baselines in simulation experiments. Website: tinyurl.com/robofume

**Keywords:** Autonomous RL, scalable robot learning, vision language models for robotics

## 1 Introduction

In many domains that involve machine learning, a widely successful paradigm for learning task-specific models is to first pre-train a general-purpose model from an existing diverse prior dataset, and then adapt the model with a small addition of task-specific data [1, 2, 3, 4, 5]. This paradigm is attractive to real-world robot learning, since collecting data on a robot is expensive, and fine-tuning an existing model on a small task-specific dataset could substantially improve the data efficiency for learning a new task. Pre-training a policy with offline reinforcement learning and then fine-tuning it with online reinforcement learning is a natural way to implement this paradigm in robotics. However, numerous challenges arise when using this recipe in practice. First, off-the-shelf robot datasets often use different objects, fixture placements, camera viewpoints, and lighting conditions compared to the local robot platform. This causes non-trivial distribution shifts between pre-training and online fine-tuning data, which makes effectively fine-tuning a robot policy difficult. Indeed, most existing works only show the benefit of the pre-train and fine-tune paradigm where the robot uses the same hardware instance in both pre-training and fine-tuning phases [6, 7]. Second, training or fine-tuning a policy in the real world often requires extensive human supervision, which includes manually resetting the environment between trials [8, 9, 10] and engineering reward functions [11, 12, 7]. In this work, our goal is to address these two challenges and develop a practical framework that enables robot fine-tuning with minimal time and human effort.

Over the past few years, there has been a lot of progress in designing efficient and autonomous reinforcement learning algorithms. However, no existing system could both utilize diverse demonstration datasets and learn with minimal human supervision, without the need for human-engineered reward functions and manual environment resets. Some works propose to reduce the need for man-

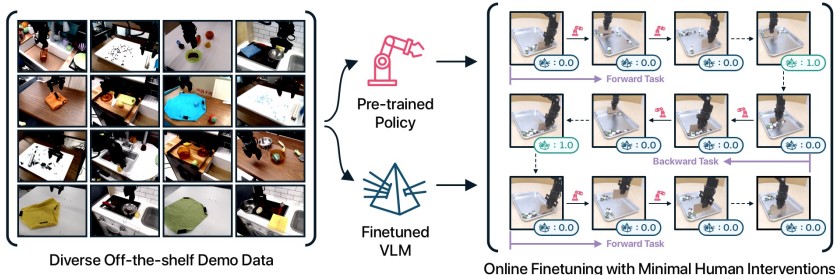

Figure 1: We propose a system that enables autonomous and efficient real-world robot learning. First, we pre-train a **multi-task policy** and fine-tune a pre-trained **Vision-Language Model (VLM) as a reward model** using diverse off-the-shelf demonstration datasets. Then, we fine-tune the **pre-trained policy** online reset-free with the **VLM reward model**.

ual environment resets using reset-free RL [11, 7, 13], where an agent alternates between running a task policy and a reset policy during training while updating both with online experience. However, these works do not leverage diverse off-the-shelf robot datasets. Recent advances in offline RL algorithms have enabled policies to leverage diverse offline data and improve further via online fine-tuning [14, 15], but these new methods have not been integrated into a system that aims at minimizing human supervision in the fine-tuning phase. There are also works that propose to eliminate the need for human-specified reward functions by learning reward prediction models [16, 17, 18, 13], but we found that many of these proposed models can be brittle when deployed in a real-world RL fine-tuning setup. In summary, although prior works have presented individual components that are vital to building a working system for efficient and human-free robot learning, it is not clear which components one should use to put together such a system and how.

We design ROBOFUME, a system that enables autonomous and efficient real-world robot learning by leveraging diverse offline datasets and online fine-tuning. Our system operates in two phases. In the pre-training phase, we assume access to a diverse prior dataset, a few task demonstrations and reset demonstrations of the target task, and a small collection of sample failure observations in the target task. From this data, we learn a language-conditioned, multi-task policy with offline RL. To cope with the distribution shift between the offline dataset and online interactions, we need an algorithm that could effectively digest diverse offline data, and display robust fine-tuning performance when placed into an environment different from those seen in the offline dataset. We find that calibrated offline RL techniques [15], by underestimating predicted values of the learned policy from offline data and correcting the scale of the learned Q-values, make sure that the pre-trained policy can effectively digest diverse offline data and continuously improve during online adaptation. To ensure the online fine-tuning phase requires minimal human feedback, we need to remove the need for reward engineering by learning a reward predictor. Our insight is to take a large vision-language model (VLM) to provide a robust pre-trained representation and fine-tune it with a small amount of in-domain data so that it is tailored for the reward classification setup. Pre-trained VLMs have already been trained on internet-scale visual and language data. This makes the model more robust to lighting and camera positioning variations than the models used in prior works. In the fine-tuning phase, a robot adapts the policy in the real world autonomously by alternating between attempting the task and attempting to reset the environment to the initial state distribution of the task. Meanwhile, the agent uses the pre-trained VLM model as a surrogate reward for updating the policy.

We evaluate our framework by pre-training it on the Bridge dataset [19] and testing it on a diverse set of real-world downstream tasks: cloth folding, cloth covering, sponge pick-and-place, placing lid on a pot, and putting a pot in a sink. We find that our system provides substantial improvements over offline-only methods with as little as 3 hours of real-world training. We perform more quantitative experiments in a simulation setup, where we illustrate that our method outperforms imitation learning and offline RL methods that either do not perform fine-tuning online or do not incorporate diverse prior data.

Our main contributions include (1) a fully autonomous system for pre-training from a prior robot dataset and fine-tuning on an unseen downstream task with a minimal number of resets and learned reward labels; (2) a method for fine-tuning pre-trained vision-language models and using them to construct a surrogate reward for downstream RL training.

## 2   Related Work

**Offline RL.** Offline RL algorithms [20, 21, 22, 23, 24, 25] provide a framework for initializing robot manipulation policies from offline demonstrations or interaction datasets. Such algorithms can also be extended to include an online fine-tuning phase after training a policy offline [26, 27, 28, 29, 30, 31, 32, 15]. Our work utilizes a recent offline RL algorithm, calibrated Q-learning (CalQL) [15], a state-of-the-art method that effectively learns from offline data and continuously improves the policy's performance online by explicitly correcting the scale of the learned Q-values. We show that integrating CalQL helps our framework effectively utilize diverse prior datasets that have large distribution shifts from real-world online interactions.

**Reset-free RL.** Training an RL policy on a real robot typically requires manual environment resets. To eliminate such need to manually reset environments, prior works have studied approaches to learn robot policies in a 'reset-free' setup. Some work [33, 34, 11, 35, 36] cast the 'reset-free' learning problem as a multi-task learning problem, observing that by learning a set of tasks where some of the tasks could reset others, an agent could then be trained to perform all of those tasks without needing manual resets. Other works [37, 38, 39, 12, 17, 13, 7] learn both a task policy and a reset policy for performing the task and resetting to the initial state distribution. Our work takes an approach between the two classes of approaches, learning a language-conditioned multi-task policy that can perform both the target task and the reset for the target task. Most of these prior works learn from scratch rather than incorporating prior data and assume that a reward function is available. ARIEL [7] combines incorporating prior data with reset-free learning but assumes a hand-crafted reward function for each environment. They also collect their own prior dataset on the same robot hardware set-up as their target task. MEDAL++ [13] learns a reward classifier with demonstration and online interaction data via adversarial training, but does not consider incorporating diverse prior data. Leveraging diverse, off-the-shelf prior demonstration datasets is desirable since these datasets are readily available to use and can help a system obtain a policy initialization for efficient fine-tuning on a target task. Our system offers an approach to both incorporate diverse prior data and improve the autonomy of the fine-tuning phase by learning a model for predicting rewards. In particular, we found out that by leveraging diverse demonstration data, our system requires only about 3 hours of training in the real world compared to 10-30 hours in MEDAL++.

**Reward learning.** Early works have studied the problem of learning a reward or cost function in imitation learning. These works leverage inverse optimal control (IOC) or inverse reinforcement learning (IRL) to extract a reward function directly from expert demonstrations [40, 41, 42]. With the advent of deep neural networks, more recent works have explored learning a reward model for an imitation learning or RL policy [43, 44, 45, 46, 47, 17, 13]. When using classifier-based reward models in reinforcement learning, RL agents can exploit the learned model by exploring states unseen during classifier training, tricking the model to output incorrect rewards. To solve such an exploitation issue, many works that learn reward models leverage adversarial learning, where a system learns a discriminator that identifies states similar to those in demonstrations as positives and those visited by the policy as negatives [44, 47, 17, 13]. However, prior work has found this training objective to be sensitive to distribution shifts between offline and online setups, such as lighting and camera view changes [48]. In this work, we fine-tune vision language models (VLM), pre-trained on internet-scale data, to construct a reward model. Large scale pre-training can learn representations that are robust to natural variations such as lighting, camera shifts and distractors [49, 16].

**Leveraging pre-trained representations as reward predictors.** Several recent works have shown positive results in utilizing pre-trained vision models [50, 16], large language models (LLMs) [51] or vision language models (VLMs) [52] as reward predictors. We tried VIP [16], a method that

126 pre-trains a visual representation for generating dense reward functions for novel robotic tasks, and
127 found it insufficient for the real-world robot fine-tuning setup. In this work, we fine-tune a pre-
128 trained VLM [53] and find that it performs most effectively as a reward model. Our proposed system
129 is flexible and can easily be adapted to use other pre-trained visual representations and VLMs.

# 3 Preliminaries

131 The goal of our method is to leverage diverse prior demonstration datasets and learn a novel target
132 task autonomously in a robot hardware instance that is distinct from the one used to collect the
133 datasets. Our method assumes access to a prior dataset $\mathcal{D}_{\text{prior}} = \cup_{j=1}^{N} \mathcal{D}_j = \cup_{j=1}^{N} \{(s_i^j, a_i^j, s_i'^j)\}_{i=1}^{K}$,
134 which consists of demonstrations of $N$ different tasks $\tau_1, \ldots \tau_k$. We assume that all demonstration
135 data uses image observations. The method will be tested on a downstream task $\tau_f$, which is different
136 from any of the prior tasks.

137 To facilitate learning on the downstream task, we also assume the availability of a small set of target
138 task demos $\mathcal{D}_f$, target task reset demos $\mathcal{D}_b$, and target task failure states $\mathcal{D}_{\odot}$. The reset demos $\mathcal{D}_b$
139 come from the reset task $\tau_b$ which resets the environment from an end state of $\tau_f$ to the initial state
140 distribution of $\tau_f$. The failure states $\mathcal{D}_{\odot}$ consist entirely of image observations that correspond to
141 unsuccessful states and are collected to aid with the VLM reward learning. In addition to all the
142 given data ($\mathcal{D}_{\text{prior}}, \mathcal{D}_f, \mathcal{D}_b, \mathcal{D}_{\odot}$), each task $\tau$ is also accompanied with a language description $l$.

# 4 ROBOFUME

144 Our work focuses on designing an efficient and scalable framework for pre-training on a diverse
145 set of prior demonstrations and autonomously fine-tuning on target tasks. Our system consists of
146 an offline pre-training phase and an online fine-tuning phase. In Section 4.1, we discuss how we
147 pre-train a language-conditioned multi-task policy on diverse data that can be fine-tuned online effi-
148 ciently. Online fine-tuning requires a reward function to label successes and failures. In Section 4.2,
149 we introduce a VLM-based classifier for providing a reward signal to the policy in the fine-tuning
150 phase. Finally, in Section 4.3, we describe how to autonomously adapt the pre-trained policy in
151 the fine-tuning phase by utilizing the VLM-based reward classifier as a reward signal and chaining
152 forward and backward behaviors to practice the task with minimal human interventions.

## 4.1 Pre-Training a Multi-Task Policy on Diverse Prior Data

154 Prior work has shown that training a policy using a conservative Q-value function is an effective
155 way to obtain a good policy from an offline dataset [22, 24]. However, fine-tuning can be critical to
156 learn competent policies as prior data may not provide sufficient coverage, especially for new tasks
157 or scenes. We leverage CalQL [15] which modifies the conservative Q-learning algorithm CQL such
158 that it enables efficient online fine-tuning by enforcing calibration on the Q-function (i.e. making
159 the Q-value of the learned policy no lower than the Monte-Carlo returns in the prior dataset). CalQL
160 allows us to improve the pre-trained policy efficiently with respect to online interactions.

161 CalQL requires the training of an actor and a critic. Since we use image observations, we addition-
162 ally train an encoder $\phi(s_{\text{img}})$ that projects the images into a lower-dimensional space before giving
163 them as inputs to the actor and critic. The encoder $\phi$ is a 4-layer CNN, and is optimized exclu-
164 sively against the critic loss. To best utilize the multi-task data, we encode task descriptions $l$ using
165 pre-trained CLIP embeddings, resulting in an embedding $z = \text{CLIP}(l)$ which is used as the task
166 representation. The policy then takes as inputs a concatenation of the encoded image observation
167 $\phi(s_{\text{img}})$, task representation $z$, and proprioceptive information $s_p$, processes the concatenated vector
168 through an MLP, and produces the output action $a$.

169 In addition to updating the policy using CalQL, we regularize policy learning with a behavior cloning
170 (BC) loss, which encourages the behaviors to stay close to the seen demonstrations. Not only does
171 this regularization improve performance of the offline pre-training, but we find that it also makes it

less likely for the autonomous fine-tuning procedure to exploit false positive rewards from the VLM reward model. The weight of the BC regularization term is chosen such that the scales of the RL loss and the BC loss are similar throughout the pre-training phase. We train the policy $\pi$ and the critic $Q$ with datasets $\mathcal{D}_{\text{prior}}, \mathcal{D}_f, \mathcal{D}_b$. After the offline learning phase, the policy and critic contain knowledge of all tasks in the prior data and the target task.

## 4.2 Fine-Tuning A Vision-Language Model for Rewards

To improve the autonomy of the policy fine-tuning phase, our agent needs to perform online fine-tuning without manually labeled or engineered reward functions. To achieve this, we propose to fine-tune off-the-shelf vision-language models as reward predictors. Leveraging existing vision-language models offers a number of benefits compared to utilizing a pre-trained visual representation or training a reward model from scratch using in-domain data: First, VLMs are trained on an Internet-scale dataset that contains diverse image and language contents. Such models possess better inductive biases and thus, can be more robust to natural shifts, such as perturbations to lighting conditions, or distractor objects that might be seen at test time. Second, since VLMs can take both visual and language information as input, they provide a natural interface for communicating the current observation and current task to the model when requesting a reward label.

We design a VLM-based reward model that takes the current observation and the task name as input and outputs a binary label of whether the current observation corresponds to a successful state or an unsuccessful state with respect to the task. Given a task name (eg. 'put green cabbage into sink'), we first use GPT4 to convert the name to a short question that could serve as a prompt to know if the task has been completed or not (eg. 'is green cabbage placed in the sink?'). Then, we pass the converted prompt to a VLM together with the current image of the environment. The VLM outputs a sparse binary reward, returning success if the 'yes' token has a higher probability than 'no' token.

We use MiniGPT4 [53] as the VLM for receiving (image, task prompt) pairs and answering whether the task has been successfully completed. We find the zero-shot performance of the pre-trained VLM to be unsatisfactory. To improve the VLM's performance for reward modeling, we fine-tune it using the prior and target task data. In particular, for every demonstration, the last 3 states are used as success states and the ground truth answer is labeled as 'Yes'; for all other states, we label the ground-truth answer as 'No'. To provide the model with more information about failed states, we collect a small dataset $\mathcal{D}_{\odot}$ of images that correspond to unsuccessful states for the forward and backward target tasks. We find in our experiments that fine-tuning leads to a more accurate reward model.

## 4.3 Autonomous Online Fine-tuning

The offline pre-training phase produces a single language-conditioned policy $\pi(\cdot|s, l)$ that can perform the target and reset tasks when provided their respective language instructions $l_f$ and $l_b$. The policy is then deployed in a hardware setup for further online fine-tuning. The outline of our pre-training and fine-tuning pipeline is presented in the Appendix in Algorithm 1.

Since we aim for a fully autonomous setup, we roll out the policy in a reset-free manner, alternating between attempting the target task $\tau_f$ with $\pi(\cdot|s, l_f)$ and the reset task $\tau_b$ with $\pi(\cdot|s, l_b)$. We use the fine-tuned VLM from the previous subsection as the sparse reward function for the RL algorithm. When the VLM predicts the task has been completed successfully, we terminate the episode and switch the language instruction for the policy to complete the other task. In addition to switching tasks upon completion as predicted by the VLM, we switch after a fixed number of timesteps (150) to ensure the robot does not become stuck in bad states. As mentioned in Section 4.1, we fine-tune the policy using CalQL with an additional BC regularization term on the critic. We find that without a BC regularization term, behaviors degrade over the course of training. By constraining the policy to stay close to the expert demonstrations from the target and reset tasks, the agent becomes less likely to exploit false-positives from the VLM reward model. We use the same fixed BC regularization weight throughout fine-tuning as we did on the offline pre-training phase. Our fine-tuning pipeline

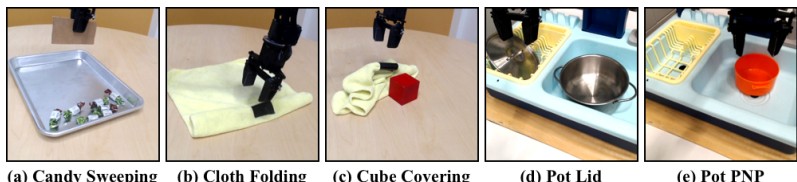

| (a) Candy Sweeping | (b) Cloth Folding | (c) Cube Covering | (d) Pot Lid | (e) Pot PNP |

Figure 2: **Illustrations of the five real-world evaluation tasks.** (a) Sweep candies to the top of the tray. (b) fold the yellow cloth. (c) cover a red wooden cube using the cloth. (d) place the lid on top of the metallic pot. (e) move the orange pot from the sink to the drying rack.

is implemented on top of the implementation of MEDAL++ [13]. Please refer to this work for more details on our training procedure.

## 5 Experiments

We design our experiments to answer the following questions: Is our method able to improve its performance through near autonomous online interactions? How does our proposed VLM reward function mechanism compare to existing alternatives? And, how does each component of ROBO-FUME or data affect the performance of our method?

### 5.1 Real Robot Experiments

**Setup.** We evaluate ROBOFUME on five different real-robot manipulation tasks. We use a WidowX 250 robotic arm with a single third-person camera (Logitech C920, resizing images to 100x100 pixels). Figure 2 shows the five tasks we fine-tune and evaluate on. Our method runs autonomously executing back and forth the target task and the reset target task for a fixed number of steps or until the VLM predicts success. For tasks involving deformable objects (the two cloth tasks) we manually reset the object to the initial forward pose every 15-25 episodes, and for the rest of the tasks we reset every 30-35 episodes. Tasks that use the kitchen-sink environment (*pot lid* and *pot pnp*) frequently experience episode interruptions when the robot arm applies more than the maximum allowed torque, for example, when close to the sink borders. All tasks use 50 forward and 50 backward demos for the target task, and fewer than 20 combined trajectories of failures. We use demos from the BridgeDataV2 [19, 54] for pre-training our language-conditioned policy, selecting approximately 1,000 trajectories with relevant behaviors per task.

**Results.** Table 1 shows the results of our method after pretraining (labeled OFFLINE) and after autonomous fine-tuning (labeled FT 30K STEPS), comparing with a behavior cloning (BC) baseline. BC trains a language-conditioned policy on all the prior and target data. After 30k steps of autonomous online interaction, our method shows relative improvement of 51% upon the pre-trained performance, and outperforms BC by 58% on an average. For pick and place tasks (*pot lid* and *pot pnp*), the fine-tuned policy was more likely to retry the action if it initially failed to grasp the object. For candy sweeping, BC and the pre-trained policy were prone to overshooting and pushing on the border of the tray after the first sweep, whereas fine-tuning the policy enabled the policy to chain multiple sweeping attempts for higher success. Additionally, we find that policies learned by ROBOFUME (both offline and after fine-tuning) to be more robust to scene distractors on the *candy sweeping* task, as reported in Table 3. The policies were trained without any distractors, but multiple objects not seen during training were placed in the background during evaluation. ROBOFUME policies retained 68% of its original performance, compared to BC which retained only 10% of its original performance. We hypothesize that BC might be more sensitive to spurious features, whereas ROBOFUME learns from more predictive features, leading to more robust policies.

### 5.2 Simulation Experiments and Ablations

We use a suite of simulated robotic manipulation environments to ablate contributions of different components of our algorithm. We test on three simulated environments used in [6]. We consider

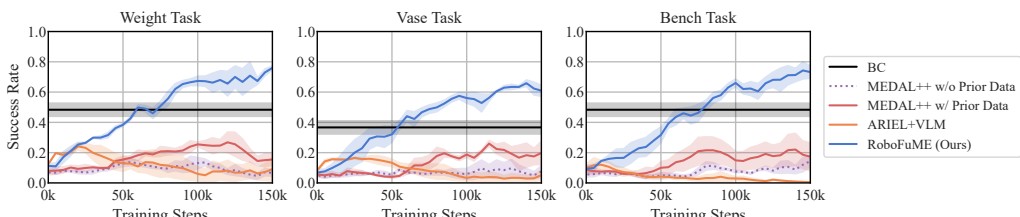

Figure 3: **Performance of our method on three simulated environments**. We report the success rate over the course of training, averaged over three seeds. Our method ROBOFUME outperforms BC, ARIEL+VLM [7], and MEDAL++ [13] consistently on all three domains.

| Task | BC | ROBOFUME (OFFLINE) | ROBOFUME (FT 30K STEPS) |
|---|---|---|---|
| Cloth Covering | 45% | 60% | **80%** |
| Cloth Folding | 60% | 70% | **85%** |
| Candy Sweeping | 31% | 47% | **66%** |
| Pot Lid | 60% | 40% | **95%** |
| Pot PNP | 45% | 35% | **55%** |

Table 1: **Real-robot results on 5 manipulation tasks.** Our method significantly improves over both offline-only and BC performance after 30k steps of online interaction (2-4 hours). For the Candy sweeping we report the average percentage of candies out of a total of 7 that are placed in the top third of the tray by the end of the evaluation. For all other tasks, we report success rate over 20 trials.

three bin-sorting tasks in which different objects (a vase, a tiny bench, and a dumbbell weight) have to be placed on the correct bin based on the language instruction, given only a sparse binary reward. We provide 10 forward and reset demonstrations for each task, 30 failure demos, and 10 demos each for 20 prior tasks that show picking and placing diverse objects on the same environment. For all methods that require online experience, we reset the environment every 1,000 environment steps, i.e. every 25 episodes of interactions. We compare our method against the following baselines: (1) *BC* behavior clones on all prior and target data; (2) *MEDAL++* learns separate forward and backward policies from target forward and backward task demonstrations and performs reset-free learning using an adversarially trained classifier as a reward signal; (3) *MEDAL++ with prior data* modifies MEDAL++ to a single language-conditioned multi-task policy and adds all prior demonstration data into the replay buffer; (4) *ARIEL+VLM* modifies ARIEL [7] to use our VLM reward models as reward signal, instead of a handcrafted ground-truth reward. The results of our simulation experiments are presented in Figure 3. In all simulation tasks, our method ROBOFUME consistently outperforms

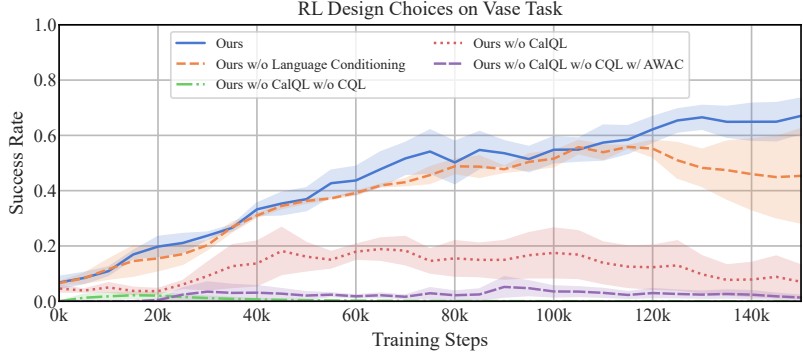

Figure 4: **Performance of our method on the Vase simulated task with different actor-critic update objectives.** Fine-tuning with CalQL is critical to obtain stable improvements on this task, as training with CQL, AWAC, or SAC yields poor performance. We also find that language-conditioned policies perform slightly better than one-hot task IDs in simulation.

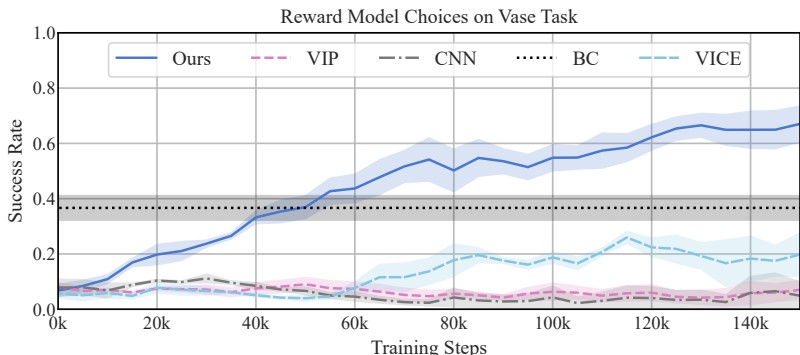

Figure 5: **Performance of our method on the simulated Vase task using different reward functions.** Our method uses a fine-tuned VLM reward function and outperforms VICE rewards, whereas CNN and VIP rewards fail to improve online.

prior methods, achieving success rates at least 20% higher than all baselines within 200k steps of online fine-tuning.

**Ablations on RL Algorithm Design Choices.** We evaluate our method trained with different critic and actor optimization procedures on the Vase simulated task, shown in Figure 4. Training with CalQL was the only method that yielded strong improvements in this task, with the other methods either failing completely or obtaining very poor performance. We find that training without the CalQL stabilizes training, while the losses for other methods would explode given the limited data.

**Ablations on Reward Models.** We compare our VLM reward function against other choices of automatic reward functions on the Vase simulated task in Figure 5. VICE [47] adversarially trains a binary classifier using positive samples from successful demonstrations, and labeling online experience as negative. We find that offline pre-training sufficiently limits the exploitation of the frozen VLM reward, outperforming VICE and thus, bypassing the need for adversarially trained reward functions. Such adversarial training can often learn to discriminate based on spurious shifts in the real world, such as lighting or scene changes, leading to instability in training outside simulation. VIP [16] trains a representation function such that the distance in representation space between the current observation and a goal image can be used to construct a dense reward function. We find that in the Vase simulated task, VIP fails to obtain good behaviors. Qualitatively, we observe VIP to be prone to false positives, which are exploited by the RL algorithm. To test the importance of the VLM large-scale pre-training compared to our fine-tuning procedure, we train a CNN classifier from scratch using the same data as we used to fine-tune the VLM, leading to unsatisfactory performance compared to fine-tuning a VLM.

## 5.3 Additional Analysis

In order to understand the performance of our method in the real robot experiments better, we performed additional analysis to examine various aspects of our framework. Please find these experiments in the Appendix.

## 6 Conclusion and Future Work

We introduced an autonomous framework that leverages existing diverse prior robot demonstration datasets and improves performance in a new robot manipulation skill by finetuning online. By combining state-of-the-art offline-to-online RL algorithms, reset-free RL, and VLM-based reward models, our framework can fine-tune efficiently and nearly autonomously. Integrating this work with new VLM models that can exhibit robust zero-shot performance on unseen manipulation tasks and improving the reset efficiency of this framework are promising directions for future research.

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

# 7 Appendix

## 7.1 Method Details

We present an overview of our system in Algorithm 1.

---

**Algorithm 1:** RoboFuME

---

Initialize agent $\mathcal{A} = \{\phi, \pi, Q\}$ and pre-trained VLM $\hat{r}$.
Initialize forward and backward tasks $\tau_f, \tau_b$.
*// Prepare data and train VLM reward classifier.*
$\mathcal{D}_{\text{prior}}, \mathcal{D}_f, \mathcal{D}_b, \mathcal{D}_{\odot} \leftarrow$ `load_data()`.
$\hat{r} \leftarrow$ `finetune_vlm`$(\hat{r}, \{\mathcal{D}_{\text{prior}}, \mathcal{D}_f, \mathcal{D}_b, \mathcal{D}_{\odot}\})$.
*// Offline pre-training phase.*
$\mathcal{A}$.`update_buffer`$(\mathcal{D}_{\text{prior}}, \mathcal{D}_f, \mathcal{D}_b)$.
**for** $t = 1$ to $T_{\textit{offline}}$ **do**
  $\mathcal{A}$.`update_params_with_calql()`.
*// Online fine-tuning phase.*
$s \leftarrow$ `env.reset()`; $l \leftarrow \tau_f$.`get_task_lang()`.
**for** $t = 1$ to $T_{\textit{online}}$ **do**
  $a \leftarrow \mathcal{A}$.`act`$(s, l)$; $s' \leftarrow$ `env.step`$(a)$.
  $\mathcal{A}$.`update_buffer`$(\{s, a, s', \hat{r}(s)\})$.
  **for** $i = 1$ to $N_{\textit{utd\_ratio}}$ **do**
    $\mathcal{A}$.`update_params_with_calql()`.
  **if** $switch$ **then**
    *// Switch task after a fixed interval.*
    $l \leftarrow$ `env.switch`$(\tau_f, \tau_b)$.`get_task_lang()`.
  **if** $interrupt$ **then**
    *// Allow occasional human intervention.*
    $s \leftarrow$ `env.reset()`; $l \leftarrow \tau_f$.`get_task_lang()`.
  **else**
    $s \leftarrow s'$.

---

## 7.2 Additional Analysis in Real Robot Experiments

| Task | FP | FN | Accuracy | Precision |
|---|---|---|---|---|
| Cloth Covering | 6.3% | 80.9% | 89.4% | 15.3% |
| Cloth Folding | 1.2% | 59.8% | 84.1% | 92.0% |
| Pot PNP | 6.1% | 81.3% | 86.9% | 24.3% |

Table 2: **VLM reward model accuracy during real robot fine-tuning.** The low false positive (FP) rate indicates that online training has minimal reward exploitation.

| Task | BC | ROBOFUME (offline) | ROBOFUME (fine-tuned @30k) |
|---|---|---|---|
| Candy Sweeping | $31\% \rightarrow 3\%$ | $47\% \rightarrow 31\%$ | $\mathbf{66\% \rightarrow 45\%}$ |

Table 3: **Robustness of learned policy to distractors.** Entries in this table show the performance of the learned policy "before" $\rightarrow$ "after" adding distractors to the scene in the candy-sweeping task. Our system learns a policy that is much more robust to the distractors.

**How Accurate is the VLM Reward?** We analyze the performance of the VLM reward over the course of fine-tuning for real-robot experiments. In Table 2, we report the false positive rate, false negative rate, accuracy, and precision metrics for the VLM reward. The metrics are computed on the data collected during fine-tuning against a hand-engineered ground truth reward. We observe that while false negative rates are high, false positive rates are low across all tasks. This asymmetry

| Task | ROBOFUME (offline) | ROBOFUME w/o Prior Data | ROBOFUME w/o Language Cond. |
|------|------|------|------|
| Candy Sweeping | 47% | 23% | 13% |

Table 4: **Evaluating effectiveness of prior data and language conditioned policies.** Results show that using prior data and using language conditioning positively affected the offline performance of our system.

is crucial for successful RL fine-tuning, as RL policies can learn poor behaviors by exploiting false positives, but labeling some successful rollouts as negatives does not necessarily impede learning.

**How Important is Diverse Prior Data and Language Conditioning?** We ablate the contribution of diverse prior data and language-conditioned policies to ROBOFUME by evaluating the offline performance on the *candy sweeping* task, reported in Table 4. When pre-training without using prior data, that is, exclusively using target data, our method is able to sweep less than half the amount of candies on average. Similarly, we find that one-hot task encodings perform substantially worse than language-conditioned policies, as the prior dataset used in real-robot training is larger and more diverse compared to the simulation experiments.

