# OpenReview forum: "Robot Fine-Tuning Made Easy: Pre-Training Rewards and Policies for Autonomous Real-World Reinforcement Learning"
_robot-learning.org/CoRL/2023/Workshop/TGR — CoRL 2023 Workshop TGR Poster_

### Official Review · Reviewer_Tecq · 2023-10-19

**Rating:** 7
**Confidence:** 3

**Review:**

This paper proposes ROBOFUME, a reset-free fine-tuning system that pre-trains a versatile manipulation policy from diverse prior experience datasets and autonomously learns a target task with minimal human input. The pipeline involves pre-training from a prior robot dataset and fine-tuning on an unseen downstream task with a minimal number of resets and learned reward labels. Both ideas of using pretraining from diverse dataset and autonomous efficient real-world finetuning can be very relevant with scalable robot learning toward generalist robot.

---

### Official Review · Reviewer_dWub · 2023-10-20

**Rating:** 8
**Confidence:** 4

**Review:**

This paper introduces an approach that leverages VLM to generate rewards and employs an inverse task to reset it. This strategy enables fine-tuning the policy in real-world scenarios with minimal human intervention needed for task resetting.

---

### Decision · Program_Chairs · 2023-10-20

**Decision:**

Accept (Poster)

**Comment:**

Great paper and closely aligned topic!